# Early interventions for post-traumatic stress following musculoskeletal trauma: protocol for a systematic review and meta-analysis

Ferozkhan Jadhakhan ![ORCID], David Evans, Deborah Falla ![ORCID]

Centre of Precision Rehabilitation for Spinal Pain, School of Sport, Exercise and Rehabilitation Sciences, College of Life and Environmental Sciences, University of Birmingham, Birmingham, B15 2TT, UK

**Correspondence to**
Professor Deborah Falla;
d.falla@bham.ac.uk

## ABSTRACT

**Introduction** Post-traumatic stress symptoms (PTSS) can be triggered following exposure to a traumatic event, such as violence, disasters, serious accidents and injury. Little is known about which interventions provide the greatest benefit for PTSS. This systematic review aims to estimate the effects of early interventions on PTSS following musculoskeletal trauma.

**Methods/analysis** Development of this review protocol was guided by the Preferred Reporting Items for Systematic Reviews and Meta-Analysis Protocols checklist. This review will include randomised controlled trials and non-randomised controlled studies evaluating the effect of early (within 3 months of a traumatic event) non-pharmacological and non-surgical interventions on PTSS in adults (aged ≥18 years). MEDLINE, PsycINFO, Embase, CINAHL, Zetoc, PROSPERO, Web of Science, PubMed and Google Scholar, as well as key journals/grey literature, will be searched from inception to 31 July 2022. Only articles published in English will be considered. Two independent reviewers will search, screen studies, extract data and assess risk of bias using the Cochrane Risk of Bias tool V.2 (RoB 2) and the Risk Of Bias in Non-randomised Studies of Interventions (ROBINS-I), respectively. Mean difference or standardised mean difference (SMD) will be extracted with accompanying 95% CIs and p values where these are reported. Group effect size will be extracted and reported. Symptoms of PTSS will be ascertained using SMDs (continuous) and diagnosis of PTSS using risk ratio (dichotomous). If possible, study results will be pooled into a meta-analysis. A narrative synthesis of the results will be presented if heterogeneity is high. The overall quality of evidence and risk of bias will be assessed using the Grading of Recommendations Assessment, Development and Evaluation, RoB 2 and ROBINS-I guidelines, respectively.

**Ethics and dissemination** Ethical approval is not required for this systematic review since data from published studies will be used. This review is expected to provide a better understanding of the effect of early intervention for PTSS following musculoskeletal trauma. Findings of this review will be disseminated in peer-reviewed publications and through national and international conferences.

**PROSPERO registration number** CRD42022333905

## STRENGTHS AND LIMITATIONS OF THIS STUDY

⇒ This systematic review protocol follows the reporting guidelines of the Preferred Reporting Items for Systematic Reviews and Meta-Analysis Protocols checklist.
⇒ This systematic review addresses a gap in the current evidence base by providing a comprehensive assessment of the effect of early non-pharmacological, non-surgical interventions for post-traumatic stress symptoms (PTSS) following musculoskeletal trauma.
⇒ A comprehensive search strategy with a wide spectrum of search terms will be used.
⇒ A meta-analysis of the results may not be possible if there is a high risk of bias among studies and methodological heterogeneity between them. If so, a narrative summary of the outcome of eligible studies may be presented in the final review.
⇒ Findings of this study may be limited by publication bias, study heterogeneity, the measurements used to assess PTSS, different psynon-pharmacological, non-surgical interventions (e.g. exercise, cognitive behavioural therapy, and massage) and the methodological quality of included studies.

## INTRODUCTION

Post-traumatic stress symptoms (PTSS) can be triggered following exposure to a traumatic event, such as violence, disasters, serious accidents and injury.[1 2] PTSS is characterised by clusters of symptoms: re-experiencing the trauma (intrusion); avoidance of stimuli related to the event; emotional numbness (dissociation); and (hyper) arousal.[3] PTSS can have a significant impact on the day-to-day life of those exposed to a traumatic event and their families,[4] affecting their social, occupational, interpersonal relationships and physical health. If these symptoms appear within 1 month of a traumatic event, and certain criteria are met, acute stress disorder (ASD) may be diagnosed.[5] In most cases, PTSS resolves on its own, but for some these symptoms may persist.[6] If these symptoms persist

beyond 1 month of the traumatic event, post-traumatic stress disorder (PTSD) may be diagnosed,[7] although this progression is not a certainty.

Musculoskeletal injuries are common and can lead to severe long-term pain and disability. They are the leading cause of severe morbidity and mortality worldwide.[8] In 2016, an estimated 146 000 deaths were attributed to road traffic accidents across the 27 countries of the European Union, equivalent to 3.2% of all deaths registered for that year.[9] Globally, road traffic accidents injured more than 50 million people and killed approximately 1.35 million people in 2016; an estimated 3700 deaths per day.[10] In England alone, an estimated 20 000 cases of major trauma occur each year resulting in 5400 deaths with many sustaining permanent disability requiring long-term care.[11] Annually in the UK, between 40 000 and 90 000 individuals are involved in a major traumatic accident, of which 50% will have sustained a musculoskeletal injury.[12 13] Some of the most common major trauma include injuries to the head, neck, spine, chest, limbs, abdomen and pelvis. In England, the most common mechanisms of injury are road traffic accidents and falls.[14] It is estimated that the annual loss in economic output as a result of major trauma is between £3.3 and £3.7 billion in the UK.[11] The annual emergency and hospital treatment for trauma care is estimated to cost the National Health Service an estimated £300–£400 million.[15]

Although ASD and PTSD are established as serious health concerns following a major physical trauma, subthreshold PTSS are also considered a serious health issue associated with significant distress and impairment.[16 17] Most people experience some PTSS immediately following a traumatic event.[18 19] For some people, symptoms develop 6 months or more after the initial traumatic event and rise above the PTSD threshold over time.[20] PTSS impacts both quality of life and healthcare resources, which is why there is a need to provide effective and timely intervention strategies.

Several non-pharmacological interventions have been shown to be effective in the management of PTSS. Several psychological treatment approaches have been studied, such as behavioural (exposure) therapy, trauma-focused cognitive behavioural therapy (CBT), supportive (psychodynamic and interpersonal) psychotherapy, brief eclectic psychotherapy and other therapies performed individually or in groups.[21] In addition, some exercise-based interventions have been developed and tested, including, aerobic exercise,[22] eye movement desensitisation and reprocessing (EMDR)[23] and prolonged exposure.[24] There have also been studies evaluating physical stimulation-based treatments such as massage[25] and acupuncture.[26]

Most psychological treatments aim to reduce PTSS by working through stressors related to the traumatic event.[27] For example, Bisson and colleagues[28] compared four 1-hour CBT sessions to no intervention (standard care only, involving no formal intervention) in individuals with evidence of PTSS. The results showed that Impact of Event Scale scores were reduced in the intervention group at 13 months, compared with the control group (adjusted mean difference=8.4, 95% CI 2.4 to 14.36). On the other hand, a systematic review and meta-analysis of multiple psychological interventions designed to treat PTSD symptoms in military personnel within 3 months of a traumatic event, found little evidence to support any interventions, although trauma-focused CBT was associated with a reduction in PTSD symptoms post-treatment compared with waitlist-control (standardised mean difference (SMD) −1.22; 95% CI −1.78 to −0.66).[29] Exercise-based interventions are purported to act on PTSS by reducing stress through the activation of endorphins[30] and potentially increasing self-efficacy.[31] Stimulation-based interventions also plausibly work through general stress reduction.[32–35] Results from a meta-analysis of four randomised controlled trials (RCTs), examining the effect of exercise (resistance training and walking) on PTSD symptoms compared with a control group, revealed that physical activity significantly reduced symptoms of PTSD, compared with control groups (Hedges' g=−0.35, 95% CI −0.63 to −0.07, p=0.02).[36]

Many of these studies have potential methodological shortcomings, justifying a robust review of empirical evidence evaluating non-pharmacological interventions in individuals with PTSS following musculoskeletal trauma. Furthermore, because of the wide range of available non-pharmacological interventions used to improve PTSS, it is important to identify which intervention(s), if any, whether alone or in combination, have the greatest effect on PTSS when applied early after (within 3 months) the traumatic event. To date, no systematic review has investigated the effect of early non-pharmacological interventions for PTSS following musculoskeletal trauma.

### Aims
This systematic review aims to:
► Synthesise evidence evaluating the effect of early non-pharmacological, non-surgical interventions on PTSS, when commenced within 3 months after a traumatic event causing musculoskeletal injuries.
► Establish which interventions are more effective at reducing PTSS severity.
► Outline implications for clinical practice.

### METHODS
This systematic review protocol follows the reporting guidelines according to the Preferred Reporting Items for Systematic Reviews and Meta-Analysis Protocols (PRISMA-P checklist)[37] (online supplemental file 1). This review protocol was registered with the International Prospective Register of Systematic Reviews (PROSPERO) on 19 May 2022 (https://www.crd.york.ac.uk/prospero/export_details_pdf.php). The planned start date for this review is 31 July 2022 and the planned end date is 30 September 2022.

### Search strategy
The following citation databases MEDLINE, PsycINFO, EMBASE, CINAHL, Zetoc, PROSPERO, Web of Science,

PubMed, Google Scholar and Cochrane central register of controlled trials as well as key journals/grey literature will be searched from inception to 31 July 2022. Due to available resources and feasibility (availability of translators within the department), only articles published in English will be considered eligible. The following keywords will be used to retrieve relevant articles: trauma, musculoskeletal, acute stress disorder, post-traumatic stress, ASD, PTSD, PTSS, therapy, intervention, cognitive behavioural, EMDR, psychological, exercise, manual therapy, massage, acupuncture. A search strategy for each database can be found in online supplemental file 2.

## Inclusion criteria

The selection criteria for inclusion/exclusion of studies will follow the Participants, Interventions, Comparators, Outcomes and Study design framework.[38]

## Population

Individuals aged ≥18 years who sustained a physically traumatic event that resulted in one or more musculoskeletal injuries. In studies with categorised aged groups, >90% of participants must be adults (≥18 years).

## Intervention

At least one non-pharmaceutical and non-surgical intervention must be applied within 3 months of the traumatic event. Types of intervention may include, for example: trauma-focused psychodynamic psychotherapy,[39] psychoeducation,[40] CBT,[41] EMDR,[42] exercise,[43] manual therapy[44] and acupuncture.[45] This review will also include any trials where a combination of two or more eligible interventions (eg, CBT and aerobic exercise) were evaluated.

## Comparator(s)/control

Comparator group(s) can be: true control (ie, no intervention provided), wait-list controls, care as usual, placebo interventions, pharmaceutical or surgical interventions, any alternative trauma-focused or non-trauma-focused psychological or physical intervention.

## Outcome measures

Outcome can be measured in terms of an improvement in or reduction of PTSS. This must be measured using a previously validated measure, which could either be the primary or secondary outcome measure of a study and reported using any statistical parameter. Studies will be included if they have used one or more validated instrument/measure to ascertain PTSS: for example, Impact of Event Scale (original, revised or abbreviated version),[46 47] Clinician Administered PTSD Scale,[48] Post-Traumatic Stress Disorder Checklist for DSM-4 (PCL, any version),[49 50] Post-Traumatic Stress Disorder Checklist for DSM-5 (PCL-5),[51] the Post-Traumatic Stress Diagnostic Scale[52] or the Acute Stress Disorder Scale,[53] or any other validated measure of PTSS.

## Study type

Any type of controlled intervention study will be eligible. Both RCTs and non-randomised controlled studies, evaluating the effect of non-pharmacological and non-surgical interventions on PTSS, will be included.

## Exclusion criteria

1. Aged <18 years.
2. Single case studies, case reports alongside any review articles, clinical guidelines, letters, editorials, studies with only abstracts and any other literature with no full-text availability and articles not published in the English language will be excluded.
3. Studies focusing solely on patients with traumatic brain injury, spinal cord injuries, burns or deliberately self-injured patients

## Measures of effect

Mean between-group difference and/or SMD will be extracted with accompanying 95% CIs and p values where this is reported. Group effect size will be extracted and reported. Symptoms of PTSS will be ascertained using SMDs (continuous) and diagnosis of PTSS using risk ratio (dichotomous).

## Preparing for eligibility screening

Before screening against eligibility criteria commences, search results retrieved from the outlined electronic databases will be assembled into a digital library and categorised by the search database using the reference management software EndNote (V.X20, Clarivate Analytics, Philadelphia, Pennsylvania, USA). Any duplicate articles will be identified and eliminated at this stage.

## Study selection

In the first reviewing round, two reviewers (FJ and DE) will independently screen titles and abstracts of articles within the digital library that potentially meet the predetermined inclusion criteria. Both reviewers will then independently select potential eligible articles for full-text screening and apply eligibility criteria to select appropriate articles for inclusion in the review. Any disagreement will be resolved by consensus. If a resolution is not reached, a third reviewer (DF) will arbitrate any disagreement over study eligibility and resolve through discussion. Table 1 shows a checklist based on study eligibility criteria to ensure that studies are correctly identified and classified appropriately. A (PRISMA-P) flowchart will be generated and will document the selection process (inclusion and exclusion) along with reasons for exclusions.

## Patient and public involvement

No patients or members of the public were directly involved in the design, writing or editing of this systematic review protocol. We will present the results of this review to our established patient and public involvement group.

**Table 1**  Eligibility criteria

| | |
|---|---|
| Study design | ▶ Randomised controlled trials.<br>▶ Non-randomised controlled studies. |
| Study characteristics | ▶ Written in English language.<br>▶ Study identified via electronic database search, grey literature, google scholar or reference lists of eligible studies.<br>▶ Full-text article available.<br>▶ Single case studies, case reports alongside any review articles, clinical guidelines, letters, editorials, studies with only abstracts and any other literature with no full-text availability and articles not published in the English language will be excluded. |
| Participants | ▶ Adults aged (≥18 years) who have sustained a physically traumatic event resulting in one or more musculoskeletal injuries.<br>▶ In studies with categorised aged group, >90% of participants must be adults (≥18 years).<br>▶ Studies focusing solely on patients with traumatic brain injury, spinal cord injuries, burns or deliberately self-injured patients. |
| Interventions | ▶ One or more non-pharmacological, non-surgical interventions (eg, psychological interventions, exercise, eye movements and/or manual therapy), commencing within 3 months after a physically traumatic event that caused musculoskeletal injuries.<br>▶ One or more comparator group. |
| Outcome measures | ▶ PTSS measured using one or more validated instruments/measures.<br>▶ Medical records to obtain participants' clinical diagnoses of PTSS. |

PTSS, post-traumatic stress symptoms.

## Data extraction (selection and coding)

Data will be managed using EndNote, V.X20 software (Clarivate Analytics). This will enable reviewers to access and share libraries, remove duplicates, review eligibility and store full texts and abstracts. Two reviewers will extract data from the included studies independently. Any disagreement over the eligibility of a study will be resolved through discussions and involve a third reviewer if needed. For missing data, up to two attempts will be made to contact study authors by email and/or phone to obtain further information. Data to be extracted will include: title of study, first author, year and country of publication, study sample size, demographic characteristics of the study sample (eg, age, gender, ethnicity, marital status and education level), description of the intervention(s), description of the comparator/control group(s), duration of the study, diagnostic tool(s) used to assess PTSS, intervention effect estimates on PTSS, proportion of eligible patients who agreed to participate, descriptors of compliance with the intervention and PTSS outcome measure status (primary or secondary). Two reviewers will independently conduct data extraction from each study using a predefined data extraction sheet. Extracted outcome data will be preintervention and postintervention mean and SD or risk ratio. Data presented as medians or alternate measures of spread will be converted to mean and SD. When only figures are presented (rather than numerical data within text), data will be extracted and analysed where possible using a software tool such as WebPlotDigitizer.[54]

## Risk of bias (quality) assessment

The Cochrane Risk of Bias tool V.2 (RoB 2)[55] will be used to assess the risk of bias of each of the included randomised trials. This tool measures the potential risk of bias across five domains; (1) randomisation; (2) deviations from intended interventions; (3) missing outcome data; (4) measurement of the outcome data; and (5) reporting bias (selective outcome reporting). The Risk Of Bias in Non-randomised Studies of Interventions (ROBINS-I)[56] tool will be used to assess the risk of bias of non-randomised studies of interventions. Two reviewers (FJ and DE) will be involved in the quality assessment and any disagreements will be resolved through a third reviewer (DF) if needed. The Grading of Recommendations Assessment, Development and Evaluation (GRADE)[57] approach will be used to assess the quality of the pooled evidence. Bias will be assessed as high, uncertain or low risk. The risk of bias judgements will be taken into account in the final consideration of treatment effect, and studies with a high risk of bias will be flagged.

## Strategy for data synthesis

Meta-analyses on reported outcomes will be performed depending on the extent of between-study variation in participant populations, interventions, study design outcomes and methodological rigour and the number of studies reporting similar effect measures using the same assessment measures. A pairwise random-effects meta-analysis will be conducted depending on the effect measures reported in the studies and similarities between individual studies, interventions and outcomes and the statistical heterogeneity, the assessment of whether genuine differences exist between results is low.[58] Heterogeneity will be evaluated using the $I^2$ statistical analysis: an $I^2$ of 50% and above is considered a high level of statistical heterogeneity.[59] Meta-analysis will be performed if heterogeneity between the studies is low ($I^2 < 50\%$).

Where possible, SMD and 95% CI will be extracted and reported as effect estimates of scales assessing PTSS. Additionally, Cohen's *d* will be extracted or calculated: effect size will be defined as small (0.0–0.2), medium (0.3–0.7) and large (>0.8). Dichotomous outcomes (ie, presence of PTSS or not) will be pooled separately and risk ratios with accompanying 95% CI extracted or calculated from raw data in exposed and comparator groups. If the level of heterogeneity and risk of bias is high between studies and pooled analysis of the studies is not possible, a narrative summary of the outcome of the selected studies will be undertaken and presented in the final review. Publication bias and small study effects (funnel plot asymmetry) will be assessed by the regression-based tests proposed by Debray and colleagues.[60] All analyses will be conducted in Stata .17.0 (Stata Corp, College Station, Texas, USA).

### Heterogeneity assessment

Assessment of the between-study heterogeneity will be statistically explored by univariate and multivariate meta-regression analyses, statistical significance will be set at ($p<0.05$). Differences in study characteristics (methodological diversity) and study populations (clinical diversity) will be further examined to explore sources of heterogeneity. Statistically significant predictors from the univariate analysis will be included in a multivariate meta-regression model, and the statistical significance will be determined at $p<0.05$. Meta-regression will be performed in Stata using the 'metareg' command.[61]

### Sub-group analyses

Subgroup analyses will be performed depending on the number of studies identified. Subgroup analyses will be performed to consider the following; (1) type of intervention and (2) type of traumatic event. Subgroup analyses may be performed to explore trauma-focused CBT, EMDR intervention or other active treatment conditions (eg, exercise or mindfulness-based interventions). The level of heterogeneity between studies will be examined using the Cochrane Q-test and quantified by $I^2$ statistical test with 95% CI. Depending on the level of heterogeneity and study characteristics, both fixed and random effect models will be used as summary effect estimates. The Mantel-Haenszel method[62] will be performed for fixed effect model if tests between study heterogeneity is not found. On the other hand, the DerSimonian and Laird method[63] will be used for random effects modelling if between-study heterogeneity is anticipated because of variations in clinical measurements (measures assessing psychological intervention), study population, type of traumatic event or regions across studies. This method incorporates a measure of the heterogeneity between studies. A minimum of two studies are generally considered sufficient to perform a meta-analysis.[64]

### Sensitivity analysis

If sufficient numbers of studies are available, we will conduct sensitivity analyses to assess the methodological quality and potential sources of heterogeneity of the included studies. Sources of variations may include assessment of psychological interventions, sampling strategy, adequate response and traumatic event. These will be stratified and separate sensitivity analyses conducted. A further analysis will be conducted excluding any studies with high risk of bias. We will explore the effect of restricting to studies with a low overall risk of bias.

### Narrative synthesis

If the level of heterogeneity is high between studies and a meta-analysis is not feasible, we will carry out a narrative summary of the outcome of the selected studies and examined in more detail and presented in the final review outlining the reasons for the results reported in each study.

### Publication bias and overall quality of the evidence

Presence of publication and small sample bias will be examined using the Egger test and inverted funnel plot technique.[65] The trim-and-fill method will be used to examine the magnitude of publication bias and estimate the overall effect of the remaining studies once studies which cause the funnel asymmetry are removed and then missing effects will be imputed until the funnel plot is symmetrical.[66] The Stata command 'metatrim' will be used to perform the non-parametric trim-and-fill technique.[67]

### Certainty of evidence

The GRADE framework[57] will be used to assess the certainty of evidence for the outcome of interest across studies. Consistent with the GRADE approach, the quality of the certainty of evidence will be assessed as high, low moderate and very low. According to the GRADE five domains, we will assess imprecision, inconsistency, indirectness, risk of bias including publication bias. Applicability of the results based on the study population will also be rated when making judgement about the quality of evidence presented in the included studies.[68] The minimum number of included studies recommended when examining publication bias is 10.[69]

## DISCUSSION

The relative strengths and weaknesses identified in the included studies will be presented in the review. One of the strengths of the proposed study is to apply a reproducible and transparent procedure to comprehensively explore the effect of early non-pharmacological intervention on PTSS, delivered within 3 months of a traumatic event that caused musculoskeletal injuries. The findings of this review will provide estimates for the effectiveness of evaluated interventions in terms of improvement in or reduction of PTSS severity. Another strength will include an in-depth search strategy structured around the main concepts being examined in this review, and adapted for each search database, and robust quality appraisal of

research-based evidence and heterogeneity assessment to evaluate studies included in this review. Some potential weaknesses are likely to include between-study heterogeneity in terms of PTSS diagnostic methods, descriptions of traumatic event, setting and/or country of study, publication bias and different non-pharmacological interventions (eg, psychoeducation, CBT, exercise, manual therapy, acupuncture and EMDR). Furthermore, PTSS commonly occurs with comorbidities and we will describe these, where available, in our review. This review will only include studies in English language, being recognised as a limitation of this review. A narrative summary of the study findings will be presented to overcome these issues if the results cannot be pooled.

## Implications of results

This systematic review will provide a synthesis of the available literature exploring the effect of early non-pharmacological, non-surgical intervention on PTSS in adults (aged ≥18 years) who have experienced a traumatic event that caused musculoskeletal injuries. Findings of this review have the potential to assist clinicians to decide which interventions, either alone or in combination, have the greatest effect on PTSS when delivered within 3 months of a physically traumatic event, and may provide an opportunity for designing or tailoring future treatment for patients experiencing PTSS.

## Ethics and dissemination

This review does not require ethical approval as only existing published data, available in scientific databases, will be used. Findings of this systematic review will be published in peer-reviewed journals and presented at conferences. Any data generated from this systematic review will be made available from the corresponding author on reasonable request.

**Contributors** FJ, DE and DF contributed equally to the conception of this protocol. FJ conceived the study design and drafted the first version of the protocol and was reviewed/revised by DE and DF. The final version was drafted by FJ. The search strategy was developed by FJ and iteration discussed with DE and DF. The final version was approved by DE and DF. The search will be performed by FJ. FJ and DE will perform initial screening for study selection. FJ and DE will collect data from the included studies and conduct quality assessment. FJ and DE will perform data analysis/synthesis. DF will ensure data extraction consistency. FJ, DE and DF drafted and critically reviewed the manuscript and approved the final version. DF is guarantor.

**Funding** This project is funded by the National Institute for Health Research (NIHR) Surgical Reconstruction and Microbiology Research Centre (SRMRC) awarded to DF. SRMRC is a research infrastructure for trauma research jointly funded by the NIHR, and the Ministry of Defence. The views expressed are those of the author(s) and not necessarily those of the NIHR or the Department of Health and Social Care.

**Competing interests** None declared.

**Patient and public involvement** Patients and/or the public were not involved in the design, or conduct, or reporting, or dissemination plans of this research.

**Patient consent for publication** Not applicable.

**Provenance and peer review** Not commissioned; externally peer reviewed.

**ORCID iDs**
Ferozkhan Jadhakhan http://orcid.org/0000-0002-4545-3703
Deborah Falla http://orcid.org/0000-0003-1689-6190

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
