## [Reviewer comments · BMJ Open]

ARTICLE DETAILS

TITLE (PROVISIONAL)	Early interventions for post-traumatic stress following musculoskeletal trauma: Protocol for a systematic review and meta-analysis
AUTHORS	Jadhakhan, Ferozkhan; Evans, David; Falla, Deborah

VERSION 1 – REVIEW

REVIEWER	Kasch, H Aarhus University, Department of Clinical Medicine
REVIEW RETURNED	13-Jul-2022

GENERAL COMMENTS	Thanks for letting me review the protocol. The overall objectives and methods applied are sound and important. The concern is how to distinguishing between musculoskeletal and CNS/PNS injuries as the substrate for PTSS. E.g head-injuries, upper neck-injuries may be more prone to the development of PTSS, not due to skeletal or muscular involvement but due to brain involvement, concussions, this is not accounted for in your presented research strategy. In a clinical setting (neurorehabilitation) it is difficult to obtain psychological treatment if there is no documented brain involvement. However, very severe injuries (multitraumatic) do qualify. It is important that these aspects are debated.
---

REVIEWER	Doody, Catherine University College Dublin, School of Public Health, Physiotherapy and Population Science
REVIEW RETURNED	15-Jul-2022

GENERAL COMMENTS	Thank you for the invitation to review this clearly described protocol for a systematic review and meta-analysis. Please see some minor comments below. In relation to the definitions of the different types of PTS, can the authors clarify the difference in terminology between PTSS and PTSD, in particular if there may be some overlap in terms of timescale since onset between these two terms (and any possible effects this may have on study selection for the review) and noting that all terms are used in the search strategy (i.e. PTS, PTSD, PTS Symptoms, acute stress disorder). Could the authors clarify further what is meant by psychological interventions? I note the search strategy interventions includes Physical Therapy, Physiotherapy and Rehabilitation The objectives of the study could be expanded somewhat to reflect
--

	the detail provided in the protocol. The methodology of the paper looks to be appropriate and well described, the review has been registered with PROSPERO, will follow PRISMA-P checklist, dates of study provided, multiple reviewers, procedures for selection of studies, data extraction, assessment, data synthesis etc. In relation to inclusion criteria/ population could the authors clarify the timeline between sustaining a physically traumatic event and when treatment would have commenced (with particular reference to PTSS definitions?)? For ease of reading consider listing exclusion and inclusion criteria for participants and study designs together. In relation to sub-group analysis could the authors clarify under 'type of intervention' what types of sub-group analysis may be carried out.
--	---

VERSION 1 – AUTHOR RESPONSE

reviewer 1: Dr. H Kasch, Aarhus University, Aarhus University Hospital Department of Neurology

Comments to the Author: Thanks for letting me review the protocol.

The overall objectives and methods applied are sound and important. The concern is how to distinguishing between musculoskeletal and CNS/PNS injuries as the substrate for PTSS. E.g head-injuries, upper neck-injuries may be more prone to the development of PTSS, not due to skeletal or muscular involvement but due to brain involvement, concussions, this is not accounted for in your presented research strategy. In a clinical setting (neurorehabilitation) it is difficult to obtain psychological treatment if there is no documented brain involvement. However, very severe injuries (multitraumatic) do qualify. It is important that these aspects are debated.

Response: This is an important point. We believe that the development of PTSS can be a precursor to PTSD, but if it occurs earlier (at least 30 days) after experiencing a traumatic event it is more likely associated with the traumatic event rather it being psychological. We have added some text to further clarify this point.

“Most people experience some PTSS immediately following a traumatic event 18-19. For some, symptoms develop six months or more after the initial traumatic event and rise above the PTSD threshold over time”. Page 6 line 170-172

“PTSS can have a significant impact on the day-to-day life of those exposed to a traumatic event and their families affecting their social, occupational, interpersonal relationships, and physical health. If these symptoms appear within one month of a traumatic event, and certain criteria are met, acute stress disorder (ASD) may be diagnosed. In most cases, PTSS resolves on its own, but for some these symptoms may persist. If these symptoms persist beyond one month of the traumatic event, post-traumatic stress disorder (PTSD) may be diagnosed, 7 although this progression is not a certainty”. Page 5 line 146-152

Reviewer 2: Dr. Catherine Doody, University College Dublin, University College Dublin

Thank you for the invitation to review this clearly described protocol for a systematic review and meta-analysis. Please see some minor comments below.

Comment: In relation to the definitions of the different types of PTS, can the authors clarify the difference in terminology between PTSS and PTSD, in particular if there may be some overlap in terms of timescale since onset between these two terms (and any possible effects this may have on study selection for the review) and noting that all terms are used in the search strategy (i.e. PTS, PTSD, PTS Symptoms, acute stress disorder).

Response: Thank you for pointing this out. We agree and have re-structured the section detailing the difference and timing of PTSS, PTSD and ASD. Page 6 line 170-172.

“Most people experience some PTSS immediately following a traumatic event. For some, symptoms develop six months or more after the initial traumatic event and rise above the PTSD threshold over time”.

“In most cases, PTSS resolves on its own, but for some these symptoms may persist. If these symptoms persist beyond one month of the traumatic event, post-traumatic stress disorder (PTSD) may be diagnosed, although this progression is not a certainty”. Page 5 line 146-152.

Comment: Could the authors clarify further what is meant by psychological interventions? I note the search strategy interventions includes Physical Therapy, Physiotherapy and Rehabilitation

Response: Thank you for pointing this out. We have added a sentence to clarify our inclusion criteria and provided examples of different types of intervention. Please see page 9 line 242-248

“A non-pharmaceutical and non-surgical intervention must be applied within 3 months of the traumatic event. Types of intervention may include, for example: trauma-focussed psychodynamic psychotherapy, psychoeducation, cognitive behavioural therapy (CBT), eye movement desensitisation and reprocessing (EMDR), exercise, manual therapy, and acupuncture. This review will also include any trials where a combination of two or more eligible interventions (e.g. CBT and aerobic exercise) were evaluated”.

Comment: The objectives of the study could be expanded somewhat to reflect the detail provided in the protocol.

Response: We thank you for this useful suggestion. We have added further details to better structure our aims. Please see page 7 line 210-215.

“Aims

This systematic review aims to:

- Synthesise evidence evaluating the effect of early non-pharmacological, non-surgical interventions on PTSS, when commenced within 3 months after a traumatic event causing musculoskeletal injuries
- Establish which interventions are more effective at reducing PTSS severity
- Outline implications for clinical practice”

Comment: The methodology of the paper looks to be appropriate and well described, the review has been registered with PROSPERO, will follow PRISMA-P checklist, dates of study provided, multiple reviewers, procedures for selection of studies, data extraction, assessment, data synthesis etc.

Response: Thank you for this comment.

Comment: In relation to inclusion criteria/ population could the authors clarify the timeline between sustaining a physically traumatic event and when treatment would have commenced (with particular reference to PTSS definitions?)?

Response: Thank you for pointing this out. We have restructured our “population” definition to include the timeline since injury occurred. Page 8 line 239-241.

“Individuals aged ≥ 18 years who sustained a physically traumatic event that resulted in one or more musculoskeletal injuries. In studies with categorised aged groups, $>90\%$ of participants must be adults (≥ 18 years)”.

Response: Thank you for pointing this out. We have listed our inclusion/exclusion criteria together as requested (Table 1).

Comment: In relation to sub-group analysis could the authors clarify under ‘type of intervention’ what types of sub-group analysis may be carried out.

Response: Agree, we have accordingly, extended our planned sub-group analyses by grouping by intervention type. Please see page 14 line 370-374.

“Sub-group analyses will be performed depending on the number of studies identified. Sub-group analyses will be performed to consider the following; [1] type of intervention; [2] type of traumatic event. Sub-group analyses may be performed to explore trauma-focused CBT (TF-CBT), EMDR intervention or other active treatment conditions (e.g. exercise or mindfulness-based interventions)”.